# Progress in Targeted Alpha-Particle-Emitting Radiopharmaceuticals as Treatments for Prostate Cancer Patients with Bone Metastases

**DOI:** 10.3390/molecules26082162

**Published:** 2021-04-09

**Authors:** Chirayu M. Patel, Thaddeus J. Wadas, Yusuke Shiozawa

**Affiliations:** 1Department of Cancer Biology and Comprehensive Cancer Center, Wake Forest University Health Sciences, Winston-Salem, NC 27157, USA; cpatel@wakehealth.edu; 2Department of Radiology, University of Iowa, Iowa City, IA 52242, USA; thaddeus-wadas@uiowa.edu

**Keywords:** bone metastases, radiopharmaceuticals, radium-223, actinium-225, targeted alpha-particle-emitting radiopharmaceuticals

## Abstract

Bone metastasis remains a major cause of death in cancer patients, and current therapies for bone metastatic disease are mainly palliative. Bone metastases arise after cancer cells have colonized the bone and co-opted the normal bone remodeling process. In addition to bone-targeted therapies (e.g., bisphosphonate and denosumab), hormone therapy, chemotherapy, external beam radiation therapy, and surgical intervention, attempts have been made to use systemic radiotherapy as a means of delivering cytocidal radiation to every bone metastatic lesion. Initially, several bone-seeking beta-minus-particle-emitting radiopharmaceuticals were incorporated into the treatment for bone metastases, but they failed to extend the overall survival in patients. However, recent clinical trials indicate that radium-223 dichloride (^223^RaCl_2_), an alpha-particle-emitting radiopharmaceutical, improves the overall survival of prostate cancer patients with bone metastases. This success has renewed interest in targeted alpha-particle therapy development for visceral and bone metastasis. This review will discuss (i) the biology of bone metastasis, especially focusing on the vicious cycle of bone metastasis, (ii) how bone remodeling has been exploited to administer systemic radiotherapies, and (iii) targeted radiotherapy development and progress in the development of targeted alpha-particle therapy for the treatment of prostate cancer bone metastasis.

## 1. Introduction

As a result of improvements in cancer research, prevention, early diagnosis, and treatment, the survival time of cancer patients with localized disease has increased. However, the prognosis for cancer patients with disseminated disease has decreased dramatically; distant metastases are responsible for 90% of all cancer-related deaths [1]. Although cancer cells may spread to any part of the body, different cancers have been observed to colonize different organs of the body at different rates and bone is a major metastatic site for several cancers. The relative incidence of bone metastasis is 65–75% in breast cancer, 65–75% in prostate cancer, 60% in thyroid cancer, 30–40% in lung cancer, 40% in bladder cancer, 20–25% in renal cell carcinoma, and 14–45% in melanoma [2]. Furthermore, the median survival time of patients with bone metastases is 19–25 months for breast cancer, 12–53 months for prostate cancer, 28 months for thyroid cancer, 6 months for lung cancer, 6 months for bladder cancer, 12 months for renal carcinoma, and 6 months for melanoma [3]. Importantly, the presence of metastatic bone disease alters the course of clinical care of patients, increases the immense physical and emotional burdens faced by patients, and augments the economic burden faced by patients and society [4,5,6,7,8,9,10,11,12,13]. For example, prostate cancer patients experiencing bone metastases incurred health care costs that were approximately $8000 more than those incurred by men without bone metastasis [14]. Interestingly, it has been demonstrated that the total medical care costs for breast cancer patients with bone metastases who experienced skeletal-related events (SREs) are nearly $50,000 greater than for those without SREs [4,7].

Since bone metastasis is one of the major causes of death of cancer patients, eradicating cancer-induced bone diseases represents one of the greatest challenges of modern health care. Thus, there is a critical need to integrate our current understanding of cancer metastasis with emerging concepts in bone biology to advance our understanding of cancer-induced bone diseases, with the goal of improving treatment strategies and clinical outcomes, while reducing the financial difficulties experienced by patients. The treatment strategies for bone metastases are somewhat unique when compared to those for other metastases. Normally, the treatment strategies for both primary and metastatic tumors are similar—targeting the tumors themselves or inducing the immune system surrounding the tumors. However, for bone metastases, the treatments target the function of the metastatic organ, which is bone (an organ that continuously remodels throughout life by coupling osteoclast and osteoblast activity, which is called bone remodeling [15]). It has been suggested that the cells involved in bone remodeling (e.g., osteoclasts, osteoblasts, and osteocytes) and bone metastatic cancer cells interact with each other, and this crosstalk between bone-related cells and bone metastatic cancer cells stimulates further bone metastatic progression, known as “the vicious cycle of bone metastases” [16]. It is therefore natural to target bone remodeling to interfere with this cycle. Indeed, bisphosphonate and denosumab, a human monoclonal anti-receptor activator of nuclear factor κB ligand (RANKL) antibody, which decreases osteoclastic activity, have been used as treatments for bone metastases [17,18]. These treatments have been effective in reducing the painful complications of bone metastases but ultimately fail to improve the overall survival of cancer patients with bone metastases [17,18]. However, recent clinical trials indicate that an alpha-particle-emitting radiopharmaceutical radium-223 dichloride (^223^RaCl_2_), which targets hydroxyapatite or osteoblastic bone metastatic lesions, improves the overall survival of prostate cancer patients with bone metastases. Importantly, to date, this is the only bone-targeted treatment modality that can prolong the survival time of cancer patients with bone metastases, although several combinations of systemic treatments (e.g., hormone therapies and chemotherapies) are known to enhance the overall survival of metastatic castration-resistant prostate cancer patients, including patients with bone metastases [19]. Although ^223^RaCl_2_’s success holds promise for alpha-particle-emitting radiopharmaceuticals for bone metastatic disease and has renewed interest in the development of these therapies [20,21,22,23,24,25,26,27,28,29,30,31,32], little is known as to the targeted treatment strategies for bone metastatic disease using alpha-particle-emitting radiopharmaceuticals.

While many excellent reviews relating to targeted radiotherapy have been published [22,33,34,35], this review will highlight (i) the biology of bone metastases by emphasizing the vicious cycle of bone metastasis, (ii) the treatment strategies for bone metastasis by mainly focusing on radiopharmaceuticals, and (iii) the future directions for targeted alpha-particle-emitting radiopharmaceutical treatment strategies in bone metastasis.

## 2. The Biology of the Vicious Cycle of Bone Metastases

Although personalized medicine strategies continue to be adopted in the clinic, the tumor phenotype at the primary site typically dictates the treatment regimen. However, these conventional treatment strategies usually fail to eradicate metastases. This leads to a more aggressive combination treatment, including chemotherapies, radiotherapies, immunotherapies, and/or targeted therapies, for cancer patients with metastases. Importantly, when considering the treatment of bone metastasis, the dynamics of bone turnover also needs to be considered during treatment planning.

Unlike other organs, bone is continuously renewed throughout life to maintain its structural integrity. Bone is composed of three parts: compact bone, trabecular bone, and bone marrow. Compact bone is a hard, solid bone tissue and forms the outside layer of bone. Trabecular bone (or spongy bone) and bone marrow are found in the inside of bones. A part of trabecular bone eventually converts into compact bone. The bone marrow is composed of two distinct stem cell lineages, cells of hematopoietic origin and those of mesenchymal origin. Hematopoietic stem cells (HSCs) give rise to all blood cell types, including macrophages that differentiate into osteoclasts, while mesenchymal stem cells (MSCs) are responsible for the generation of stromal cells, osteoblasts, and osteocytes [36]. Additionally, these cells of mesenchymal origin are crucial for trabecular bone development. Interestingly, these two cell lineages interact with each other to maintain each other’s functions. For example, osteoblasts serve as the microenvironment for HSCs, or the HSC niche [37,38,39,40,41,42], while HSCs support osteoblastic differentiation to establish the HSC niche [43]. Another example is that osteoblasts are responsible for the activation of osteoclasts [44], while these activated osteoclasts resorb the bone matrix to create the space for osteoblasts to form new bone [45]. As a result, compact bone, trabecular bone, and bone marrow crosstalk to maintain healthy bone development. This process, called bone remodeling, is a delicate balance that is often exploited by cancer cells that successfully colonize the bone [46].

Since Paget’s seed and soil theory was proposed over a century ago [47], efforts have been made to understand why particular cancers prefer specific organs over others. Although blood flow and anatomical structure are considered as among the major contributing factors to the development of bone metastasis [48,49], proper physiological mechanisms of why particular types of cancers disseminate to the bone have not been fully uncovered. Additionally, recent evidence suggests that the establishment of organ-specific metastases is the result of not only the passive reception of circulating tumor cells through blood flow and anatomical structure but also the fact that the bone microenvironment selectively and actively recruits these circulating tumor cells [50]. This indicates that the interactions between the microenvironment of the bone marrow and bone metastatic cancer cells are crucial for the bone metastatic progression process.

Once these tumor cells have effectively seeded on the bone marrow, they begin to proliferate and interact with the cells involved in bone remodeling (e.g., osteoclasts, osteoblasts, and osteocytes) through paracrine and juxtacrine signaling events. This interaction creates an imbalance in the normal bone remodeling process in what has been termed the vicious cycle of bone metastases (Figure 1) [51]. Recent studies have revealed that bone metastatic prostate cancer cells hijack the interaction between the cells of hematopoietic and mesenchymal lineages, which is important for maintaining healthy bone remodeling, to establish metastatic growth within the marrow. For example, bone metastatic prostate cancer cells target the osteoblastic HSC niche during their dissemination to the bone and compete for occupancy of the HSC niche [52,53]. These observations are consistent with previous research that has demonstrated that metastatic colonization is frequently observed in bones that contain red marrow, where blood cell formation and bone formation are active (e.g., the axial skeleton, vertebrae, ribs, and the pelvis) [54].

Additionally, once the vicious cycle has been initiated, bone metastases may present as osteolytic (bone destructive), osteoblastic (bone forming), or mixed metastases, depending on how the infiltrating cancer cells have exploited the normal bone remodeling mechanisms. Although there are osteolytic- and osteoblastic-only bone metastases, most metastases arising from solid tumors have a heterogenous phenotype [55]. Osteolytic bone metastatic lesions are characterized by the destruction of normal bone formation, primarily mediated by the hyper-activation of osteoclasts (multinucleated cells that resorb the bone matrix and develop from a monocyte–macrophage lineage) [56]. In this destructive mechanism, bone metastatic cells induce the release of parathyroid-hormone-related peptide (PTHrP), which is known to activate osteoclasts and induce bone resorption [57,58]. Conversely, osteoblastic bone metastatic lesions are characterized by the deposition of new bone. Although the mechanisms of osteoblastic bone metastases are still poorly understood, they are believed to occur through the hyper-activation of osteoblasts [59]. However, the newly formed bone matrix is poorly organized, weak, and fragile [60,61]. This leads to a lack of mechanical strength and frequent fracture [60,61]. Moreover, as osteoblasts continue to proliferate, they can inadvertently cause increased bone resorption, since they release cytokines, such as RANKL, which stimulates osteoclast differentiation and activation [59,60,62].

Taken together, the bone marrow microenvironment, especially where active hematopoiesis and bone remodeling take place, may play a crucial role in the establishment and development of bone metastases, and therefore bone-marrow-microenvironment-targeting strategies have been used to treat bone metastatic disease.

## 3. Treatment of Bone Metastases

Current treatment options for bone metastasis include agents targeting bone remodeling (e.g., bisphosphonate and denosumab), hormone therapy, chemotherapy, external beam radiation therapy (EBRT), systemic radiotherapy, and surgical intervention [3,63,64]. Often, more than one intervention is used to suppress bone metastatic growth and maintain a patient’s quality of life [65]. Although EBRT, bisphosphonate, and denosumab can reduce the onset of the painful complications of bone metastasis [17,18,66,67,68,69,70,71], delivering EBRT to every bone metastatic lesion is not practical and bisphosphonate and denosumab fail to improve the overall survival of bone metastatic patients (since these therapies mainly target bone remodeling but not cancer cells within the bone). Because of these limitations, bone metastasis is currently considered a hard-to-treat disease. Therefore, significant effort has been expended to develop ways to use the systemic delivery of therapeutic radionuclides to every bone metastatic lesion so that bone metastatic cancer cells can be exposed to the cytocidal effects of radiation.

### 3.1. Radiopharmaceuticals for Bone Metastases

Radionuclides derive their cytotoxicity from the particles they release during radioactive decay, which has been considered a systemic radiotherapy for bone metastasis. These particles include auger electrons (e^−^), beta minus (β^−^) particles, and alpha (α^++^) particles (Table 1) [72].

Auger electrons are released from an atom that undergoes electron capture or internal conversion. These particles have low energy, travel a maximum distance of a micron in tissue and have moderately high linear energy transfer (LET), which allows for many destructive ionization events along the path traveled by the particles [73]. These properties require that the emitting atom be localized in close proximity to the cancer cell’s DNA to have a therapeutic effect [74,75].

Beta-minus-particle-emitting radionuclides decay by β^−^ emission. These negatively charged electrons have widely varying energies and path lengths, which may range from 0.05 keV to 2.3 MeV and from 0.05 mm to 12 mm, respectively. As these particles travel, they exhibit a lower LET than auger electrons. This results in few ionization events along the particles’ path. Their cytocidal effect is believed to occur through the particles’ ability to generate reactive oxygen species (ROS). In turn, these ROS generate single-stranded DNA breaks within the cancer cell. High concentrations of these β^−^-particles are required to create enough single-stranded DNA breaks to overwhelm DNA damage repair mechanisms and yield a therapeutic benefit.

Alpha-particle-emitting radionuclides are, by far, the most cytocidal radionuclides that have been used as systemic radiotherapy [76,77]. These particles have mass and charge equivalent to those of a helium nucleus. Alpha-particles are emitted with an energy range of 5–9 MeV, travel short distances (equivalent to no more than 10 cell diameters), and have an LET that surpasses the LET of auger electrons to ensure a significant number of ionization events along the alpha-particle’s path through tissue. These properties make the therapeutic efficacy of these radionuclides less dependent on chemo- and radio-resistance mechanisms or hypoxia. Although their cytotoxic properties have been recognized for decades, they are only now achieving clinical translation due to successful research initiatives that have explored how to efficiently attach these α^++^-emitting radionuclides to targeting ligands such as antibodies and peptides for effective delivery to cancer cells.

Historically, the Food and Drug Administration (FDA) has approved beta-minus-particle-emitting radiopharmaceuticals as adjunct treatments for bone metastases because of their chemical properties that give them an affinity for the bone matrix [78,79,80,81]. These radiotherapies are either administered as an ionic salt solution or chelated to bone-seeking ligands, such as hydroxyethylidene diphosphonic acid (HEDP) or tetramethylene phosphonic acid (EDTMP) (Figure 2A); a brief review of each radionuclide is outlined below.

Phosphorus-32 and strontium-89: Phosphorus-32 (^32^P: t_1/2_ = 14.3 d; E_βmax_ = 1.71 MeV; maximum penetration in tissue (average) = 8 mm (3 mm)) is produced by the irradiation of sulfur-32 (^32^S) through neutron capture and is distributed as ^32^P-orthophosphate. It has been investigated as a bone-pain-palliating radiopharmaceutical for more than 50 years [82]. After injection, approximately 90% of the injected dose is known to be incorporated at sites of bone remodeling due to the radiopharmaceutical’s affinity for the hydroxyapatite within the bone matrix. In several studies, palliative response rates approached 80% [83,84,85] and were observed to last more than a year in patients who received multiple doses of the radiopharmaceutical. Unfortunately, due to the long particle path length and relatively high energy of the β^−^-particle, pancytopenia was observed in most patients, resulting in disfavor of its clinical use [86,87]. Strontium-89 (^89^Sr: t_1/2_ = 50.5 d; E_βmax_ = 1.46 MeV; maximum penetration in tissue (average) = 6 mm (2.4 mm)) is produced by nuclear fission or through neutron capture and is distributed as ^89^SrCl_2_. Since ^89^Sr^2+^ ions exhibit similar chemical properties to Ca^2+^ ions, they are readily incorporated into the hydroxyapatite where active bone remodeling is occurring due to metastatic disease [78]. Typically, overall response rates, as defined by a reduction in bone pain, for patients receiving ^89^Sr therapy approached 80%, with the palliative effects lasting approximately 15 months [87,88]. Although significant changes in red blood cell counts are not observed with this therapy, a decrease in white blood cell counts and platelets has been observed in 80% of patients [89,90,91,92], but these toxicities are resolved within 4 months of treatment cessation.

Rhenium-186 and Rhenium-188: Rhenium-186 (^186^Re: t_1/2_ = 3.8 d; E_βmax_ = 1.07 MeV; maximum penetration in tissue (average) = 4.5 mm (1.1 mm)) is routinely produced in nuclear reactors by direct neutron activation of metallic-enriched ^185^Re. Rhenium-186 decays with a maximum beta energy of 1.07 MeV and a low abundance 137 keV gamma emission. It has a physical half-life of 89.3 h. Because of its maximum beta energy and particle path length, efforts have been made to determine its ability to treat bone metastasis, but since its chemical properties are not similar to those of calcium, it must be chelated to a bone-seeking ligand before injection. In the comparative study between the skeletal and soft-tissue uptake of ^186^Re-HEDP (0.13 GBq) and ^153^Sm-EDTMP (37 MBq/kg-body weight) in patients with confirmed bone metastasis, significantly less bone uptake was observed in patients receiving ^186^Re-HEDP therapy, while soft-tissue retention was comparable for both radiopharmaceuticals [78]. Despite this fact, several studies have demonstrated that more than 80% of patients receiving ^186^Re therapy reported an improvement in their bone pain [93,94,95,96,97]. Similar to other agents, thrombocytopenia was observed to be a major but reversible side effect of treatment [98,99]. Rhenium-188 (^188^Re: t_1/2_ = 0.7 d; E_βmax_ = 2.12 MeV; maximum penetration in tissue (average) = 10.4 mm (3.1 mm)), although not approved for clinical use, has also been investigated as an agent for bone pain management, since it can be produced and shipped to clinical sites as a ^188^W/^188^Re generator [100]. When chelated to HEDP, it demonstrated therapeutic efficacy, and repeated administrations improved progression-free and overall survival [101]. Additionally, it was observed that only 63% of the patients receiving ^188^Re-HEDP therapy reported grade I thrombocytopenia, 3% reported Grade II thrombocytopenia, and 3% of the patients reported grade I leukopenia, which were comparable for patients treated with ^186^Re-HEDP or ^153^Sm-EDTMP [101]. Furthermore, platelet and leukocyte counts returned to pretreatment levels within 3 months, suggesting that these toxicities are clinically manageable.

Samarium-153: Samarium-153 (^153^Sm: t_1/2_ = 1.9 d; E_βmax_ = 0.81 MeV; maximum penetration in tissue (average) = 2.5 mm (0.6 mm)), which is produced by the neutron irradiation of a samarium-152 target, has a half-life of 46.3 h and a range in bone of 1.7 mm [102]. To be effective, however, ^153^Sm needs to be chelated to a ligand such as EDTMP, which like other phosphonic acid complexes has a high affinity for skeletal tissue while being rapidly cleared from the blood and local soft tissue. This radiopharmaceutical has shown clinical benefit in several clinical trials, with the majority of patients experiencing pain relief [103]. In one study involving 100 patients with confirmed metastatic disease, subjects received 18–37 MBq/kg of ^153^Sm-EDTMP and were followed for several weeks after treatment [102]. Nearly 70% of the patients experienced symptomatic relief of their bone pain. Myelotoxicity was the main side effect of treatment, with nearly 90% of the subjects experiencing grade II thrombocytopenia or leukopenia, which resolved approximately 2 months after therapy. While several patients experienced grade III/IV myelotoxicity, these patients were observed to have a depleted hematopoietic reserve because of previous therapeutic interventions.

Lutetium-177: Lutetium-177 (^177^Lu: t_1/2_ = 6.7 d; E_βmax_ = 0.50 MeV; maximum penetration in tissue (average) = 2.2 mm (0.67 mm)) is produced by neutron irradiation of ^176^Lu or ^176^Yb targets. Although several cancer-targeting agents, such as Lutathera^®^ and ^117^Lu-PSMA-617, have been used to deliver targeted radiation therapy to cancer cells expressing specific cancer cell antigens [104], additional studies have sought to use the radioactive decay properties of ^177^Lu for treating bone metastasis. Several groups have investigated the utility of ^177^Lu-EDTMP as a bone-seeking radiopharmaceutical since the energy of the emitted β^−^ is low enough to reduce bone marrow suppression, which has consistently been a major disadvantage of other β^−^-emitting bone-seeking agents [105,106]. Recently, the use of low- and high-dose ^177^Lu-EDTMP was examined in a phase II study in patients with metastatic breast or prostate cancer [107]. In this study, subjects received either a low (0.13 GBq) or a high (0.26 GBq) dose of ^177^Lu-EDTMP and were followed for 16 weeks post-treatment. Based upon pain assessment scores, the overall response rate was 86%. Additionally, treatment-related toxicity was evaluated in all patients. Grade I/II hematological toxicities were observed in 34% of the subjects, while grade III/IV toxicities were observed in 23% of the study participants. However, the observed toxicities were not significantly different in either the low- or the high-dose cohort, and this finding prompted the authors to conclude that ^177^Lu-EDTMP is safe and effective as a treatment for bone pain palliation. More recently, several groups investigated whether using a bisphosphonate to deliver the cytocidal radiation released by ^177^Lu to the metastatic deposit in bone would provide greater relief. While many bisphosphonates were studied, zoledronic acid was chosen for further study since it had significantly higher incorporation into hydroxyapatite [108,109]. Consequently, several groups studied ^177^Lu-DOTA-zoledronic acid (^177^Lu-DOTA-ZOL) and demonstrated that it has pharmacokinetics comparable to those of ^177^Lu-EDTMP [110,111]. These studies also revealed higher absorption of the dose by the trabecular bone in patients who received ^177^Lu-DOTA-ZOL vs. ^177^Lu-EDTMP, but the dose absorbed by critical organs was much lower for the former radiopharmaceutical. Recently, the safety and efficacy of ^177^Lu-DOTA-ZOL in 40 patients with metastatic bone disease was evaluated [112]. Eligible subjects received 0.13 GBq of ^177^Lu-DOAT-ZOL at monthly intervals and then were monitored for a 12-week period after therapy. Based upon criteria such as pain palliation, an overall response rate of 90% was observed. However, unlike patients receiving ^177^Lu-EDTMP, grade III/IV hematological toxicities were not observed, although several patients did experience grade II anemia. Moreover, renal toxicity and hyper-calcemia, which can be complications of bisphosphonate administration, were not observed in study participants. Interestingly, pain palliation was observed within 7 days of treatment, with relief lasting 10 months, which is more than twice that experienced by patients receiving ^177^Lu-EDTMP.

Not only beta-minus-particle-emitting radiopharmaceuticals but also alpha-particle-emitting radiopharmaceuticals have been used as a treatment modality for bone metastatic disease. Radium-223 dichloride (^223^RaCl_2_: T_1/2_ = 11.4 d; Eα_max_ = 6–7 MeV) is a water-soluble salt that was approved nearly a decade ago by the FDA for the treatment of bone metastasis associated with metastatic prostate cancer. Similar to calcium ions, this alkaline earth ion accumulates in bone, where it decays through seven daughter radionuclides, while releasing four α^++^-particles and two β^−^-particles, generating approximately 30 MeV of total kinetic energy, which is deposited in the surrounding bone cancer microenvironment [67,68,69,78,113]. This large energy deposition is believed to generate irreparable double-stranded DNA breaks within the DNA of tumor cells that have localized in the bone, causing cancer cell death [114,115,116,117]. This strategy has yielded clinical success. For example, in the ALSYMPCA trial (NCT00699751), which was a randomized, double-blind, placebo-controlled phase III trial that investigated the role of ^223^RaCl_2_ in metastatic castration-resistant prostate cancer patients (*n* = 921), the median overall survival of patients treated with ^223^RaCl_2_ (*n* = 614, 14.9 months) significantly improved compared with those treated with a placebo (*n* = 307, 11.3 months) (hazard ratio, 0.70; 95% confidence interval, 0.58 to 0.83; *p* < 0.001) [66]. Moreover, patients treated with ^223^RaCl_2_ experienced decreased pain levels that correlated with increased overall survival [118]. Diarrhea, nausea, and vomiting were the most common side effects reported by patients. Additionally, pancytopenia was observed in many patients with grade III/IV thrombocytopenia, and neutropenia was observed in patients with compromised bone marrow function [119]. Since those initial trials, the use of ^223^RaCl_2_ therapy in combination with other agents using a strategy that has proven successful with non-radioactive therapeutics has been sought [120]. For example, a post hoc exploratory analysis of an international, early access, open-label, single-arm phase IIIb trial of the testing efficacy of ^223^RaCl_2_ in metastatic castration-resistant prostate cancer patients following the ALSYMPCA trial revealed that a combination of ^223^RaCl_2_ and androgen deprivation therapies (ADTs; abiraterone or enzalutamide) extended the patients’ overall survival compared to therapy with ^223^RaCl_2_ alone [118]. However, a recent multinational, multicenter, randomized, double-blind, placebo-controlled phase III trial of the combination of abiraterone and ^223^RaCl_2_ in patients with metastatic castration-resistant prostate cancer (NCT02043678) not only failed to improve skeletal event-free survival but also increased the frequency of bone fractures compared to the placebo [121]. These contradictory results suggest that additional prospective studies are needed and caution is warranted when choosing an effective combination strategy that involves the use of ADTs and ^223^RaCl_2_ therapy for treating prostate cancer patients with bone metastasis.

### 3.2. Targeted Radiopharmaceuticals for Bone Metastases

To date, the strategy for integrating systemic radiotherapies into the treatment plans of cancer patients with bone metastases has relied on bone metabolism to target bone metastatic disease. While successful, the results of this strategy have been primarily palliative, since the bone metastatic cancer cells are not specifically targeted by the radiopharmaceutical. As a result, efforts are being made to expand this strategy by developing agents that can selectively target biomarkers that are over-expressed on the cancer cells within the bone metastatic microenvironment, with the goal of improving therapeutic efficacy and patient outcomes. For example, one such strategy that has been successful in the treatment of metastatic prostate cancer has been the development of radiotherapies that specifically target prostate-specific membrane antigen (PSMA) [122]. PSMA is known to be highly expressed on prostate cancer cells at the primary tumor and within visceral and bone metastases [123]. PSMA-617, a small molecule that binds to PSMA with high affinity, was designed to target PSMA-expressing prostate cancer cells (Figure 2B) [124]. In a recent single-arm, single-center, phase II trial, ^177^Lu-PSMA-617 was administered to men with metastatic castration-resistant prostate cancer (*n* =30) [125]. After four cycles of radiotherapy, 57% of these patients experienced minimal toxicity and achieved a greater-than-50% reduction in prostate-specific antigen (PSA) levels, which is one of the clinical surrogate markers for prostate cancer treatment response [125]. Based on these promising results, an international prospective open-label randomized phase III trial comparing the treatment efficacy between ^177^Lu-PSMA-617 and the best standard of care in men with metastatic castration-resistant prostate cancer (VISION trial, NCT03511664) is currently underway [126], and as of 23 March 2021, the initial result that ^177^Lu-PSMA-617 significantly improves the overall survival and radiographic progression-free survival of PSMA-positive metastatic castration-resistant prostate cancer patients was announced (https://www.novartis.com/news/media-releases/novartis-announces-positive-result-phase-iii-study-radioligand-therapy-177lu-psma-617-patients-advanced-prostate-cancer, (accessed on 1 April 2021)).

Despite the promise of ^177^Lu-PSMA-617 therapy, approximately 30% of patients do not respond to this treatment [127]. These observations have led to the investigation of the use of ^225^Ac-PSMA-617 as an alternative alpha-particle-emitting radiotherapy for refractory patients [127,128,129]. Actinium-225 (^225^Ac: t_1/2_ = 10 d; Eα_max_ = 6–8 MeV) is a radioactive metal of the actinide series. Similar to ^223^RaCl_2_, ^225^Ac has a relatively long half-life and emits four α^++^- and two β^−^-particles per nuclear decay. However, unlike ^223^RaCl_2_, ^225^Ac can be linked to a variety of targeting ligands [130,131,132,133,134,135,136,137]. In the early stage of the development of ^225^Ac-PSMA-617, two patients received this therapy [127]. The first patient exhausted conventional chemotherapies, bone remodeling therapy, ADTs, and six cycles of ^223^RaCl_2_ therapy. The patient received three cycles of approximately 10 MBq (100 kBq/kg body weight) of ^225^Ac-PSMA-617. After 8 weeks, all visible visceral and bone metastases had decreased to a size that was below the limit of detection of clinical imaging scanners, while PSA levels had decreased from 3000 to 0.26 ng/mL. Similar to the first patient, the second patient was refractory to conventional chemotherapies and ADTs and failed to respond to several cycles of ^177^Lu-PSMA-617 therapy. This patient received three cycles of ^225^Ac-PSMA-617 therapy (100 kBq/kg body weight) at bimonthly intervals. After completing the three cycles, the patient experienced complete remission, as indicated by the results of the restaging PSMA-PET/CT scans (Figure 3). Similar to the first patient, no relevant hematological toxicities were observed, although both subjects did experience moderate xerostomia.

In additional studies by this same group, chemotherapy-naive prostate cancer patients with extensive bone and lymph node metastases were selected to receive ^225^Ac-PSMA-617 therapy [138]. All patients (*n* = 17) had failed prior treatments, which included radical prostatectomy, EBRT, ADT, brachytherapy, and ^177^Lu-PSMA-617 therapy. Patients received between two and six cycles of ^225^Ac-PSMA-617, with an average administered activity range between 4 and 13 MBq. Following treatment, 82% of the patients experienced a ≥90% reduction in PSA levels. Moreover, PSMA PET/CT revealed that 65% of the treated patients demonstrated a complete absence of metastatic lesions at the sites of metastases, previously identified on baseline imaging scans [138]. Similar to the initial two patients, no acute toxicities were experienced by any of the surviving patients, though all patients did experience xerostomia, which prompted additional investigations into the most efficient ways to salvage salivary gland function [139,140].

While a recent report of a patient achieving remission lasting longer than 5 years was published in the literature [141], these results should be tempered by recent observations that 17% of patients, many with bone metastasis, do not respond to ^225^Ac-PSMA-617 therapy despite sufficient tumor uptake on PSMA-PET/CT. Investigations using CT-guided biopsy and targeted next-generation sequencing have revealed a total of 15 mutations in DNA-damage-repair-associated genes, which suggests that patients harboring these mutations may need a treatment strategy that combines ^225^Ac-PSMA-617 with chemotherapies that target DNA damage repair pathways to achieve maximum clinical benefit [142].

Although ^225^Ac-PSMA-617 is a promising treatment agent for metastatic castration-resistant prostate cancer, PSMA-targeted agents (e.g., antibody, nanobody, small molecule) labeled with other alpha-particle-emitting radiopharmaceuticals have also been explored in both preclinical and clinical settings (Table 2). Since PSMA is expressed by not only bone metastatic prostate cancer cells but also those at the primary site and soft-tissue metastatic sites, this strategy is not specific for bone metastases. However, a majority of the patients with metastatic castration-resistant prostate cancer experience bone metastases during the course of disease progression. Therefore, the development of PSMA-targeted alpha-particle-emitting radiopharmaceuticals is crucial if eradicating bone metastatic prostate cancer is to be an achievable clinical goal. Further studies, including the assessments of their clinical efficacy and safety, are clearly warranted.

## 4. Conclusions

Bone metastasis creates enormous physical, emotional, and financial burdens for cancer patients and society; it is a significant contributor to cancer mortality rates. Systemically delivered radionuclides have been explored as potential therapies for bone metastasis. Initially, β^−^-emitting radionuclides were used because they could be delivered easily to sites of active bone remodeling. However, their efficacy has been limited to pain mitigation. Recently, systemically delivered ^223^RaCl_2_, which decays through α^++^-particle emission, has been observed to extend the overall survival of men with metastatic prostate cancer. These results have renewed interest in alpha-particle-emitting radiotherapies, and new molecularly targeted strategies to deliver this highly cytotoxic radiation directly to bone metastatic cancer cells within the bone marrow microenvironment are being investigated. Using this strategy, ^225^Ac-PSMA-617, which has been an effective treatment for men with metastatic prostate cancer, which was previously believed to be refractory to conventional chemotherapies and standard of care radiotherapy, has been developed. While further clinical evaluation is warranted, current data suggest that targeted alpha-particle-emitting radiopharmaceuticals used alone or in concert with current standard treatments may lead to much-needed improvements in the clinical management of cancer bone metastasis.

## Figures and Tables

**Figure 1 molecules-26-02162-f001:**
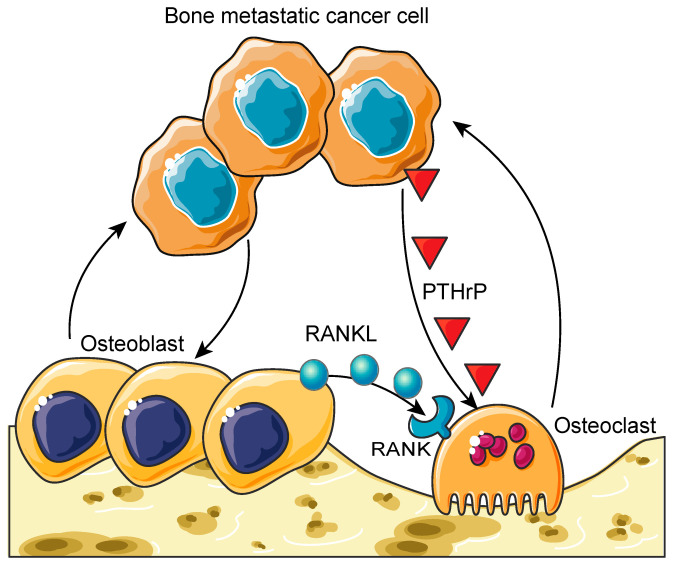
The vicious cycle of bone metastases. Bone metastatic cancer cells hijack the healthy bone remodeling process to create a suitable microenvironment for them to grow. Cancer cells induce hyper-osteoclastogenesis by activating osteoclasts through the secretion of parathyroid-hormone-related peptide (PTHrP). This process leads to osteolytic bone lesions and provides bone metastatic cancer cells more space to grow. On the other hand, cancer cells can over-activate osteoblasts, resulting in osteoblastic bone lesions. These hyper-activated osteoblasts also stimulate osteoclastogenesis through the receptor activator of the nuclear factor κB ligand (RANKL) (secreted from osteoblasts)/RANK (expressed on osteoclasts) axis. Furthermore, these hyper-activated osteoblasts and osteoclasts enhance the growth and survival of bone metastatic cancer cells. This process is called the vicious cycle of bone metastasis. Graphics adapted from Smart Servier Medical Art (https://smart.servier.com/, accessed on 18 March 2021).

**Figure 2 molecules-26-02162-f002:**
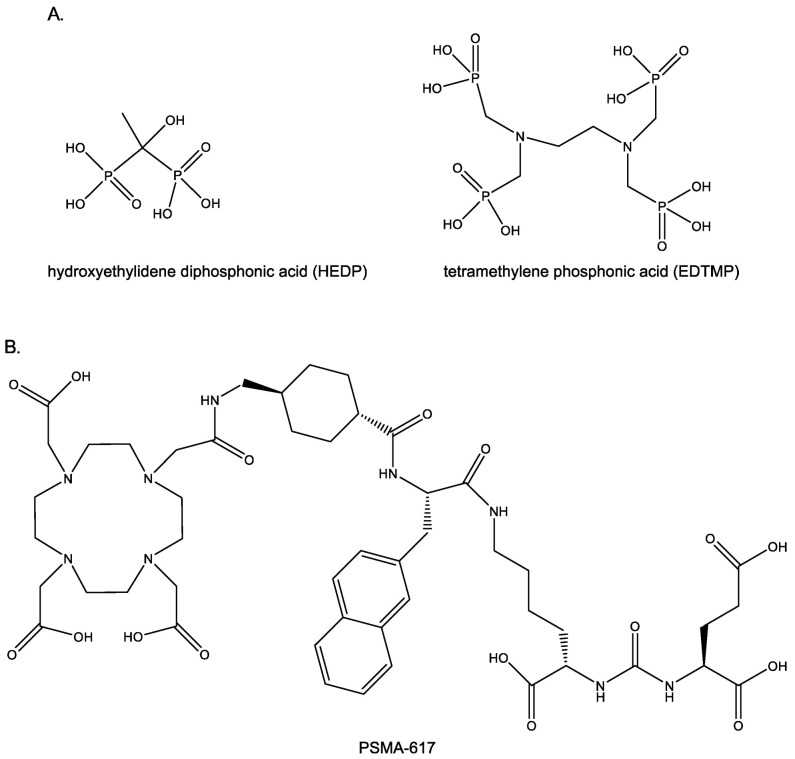
Several ligands used to deliver alpha-particle-emitting and beta-minus-particle-emitting radionuclides to bone metastasis. While ligands such as (**A**) hydroxyethylidene diphosphonic acid (HEDP) and tetramethylene phosphonic acid (EDTMP) target bone remodeling to deliver therapeutic radiation to bone metastases, ligands like (**B**) prostate-specific membrane antigen (PSMA)-617 target prostate cancer cell biomarkers.

**Figure 3 molecules-26-02162-f003:**
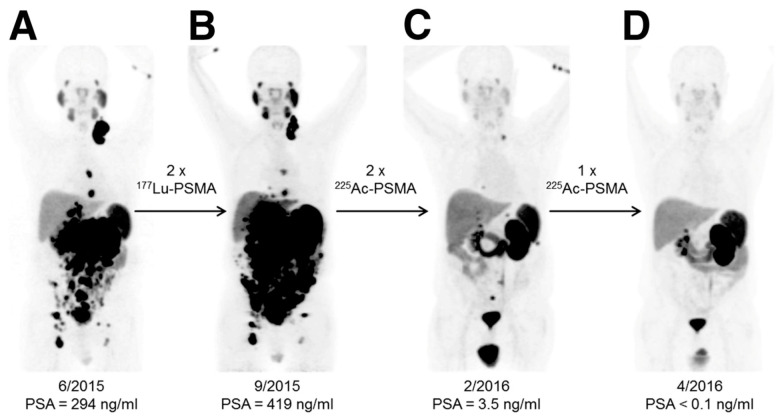
An example of the effective response of a prostate cancer patient to ^225^Ac-PSMA-617 treatment. This research was originally published in the *Journal of Nuclear Medicine* [127]. Kratochwil C, Bruchertseifer F, Giesel FL, Weis M, Verburg FA, Mottaghy F, Kopka K, Apostolidis C, Haberkorn U, and Morgenstern A. 225Ac-PSMA-617 for PSMA-Targeted α-Radiation Therapy of Metastatic Castration-Resistant Prostate Cancer. J Nucl Med. 2016;57(12):1941-1944. © SNMMI. ^68^Ga-PSMA-11 PET/CT scans of patient B in this manuscript (a prostate cancer patient who presented with peritoneal carcinomatosis and liver metastases that were progressive under ^177^Lu-PSMA-617 therapy). In comparison to the initial tumor spread (**A**), restaging after 2 cycles of β-emitting ^177^Lu-PSMA-617 presented progression (**B**). In contrast, restaging after second (**C**) and third (**D**) cycles of α-emitting ^225^Ac-PSMA-617 presented an impressive response. Adapted from ref. [127].

**Table 1 molecules-26-02162-t001:** General characteristics of therapeutic radionuclides.

Decay	Particle	Maximum Particle Range (mm)	Maximum Particle Energy (MeV)	Linear Energy Transfer (kEV/µm)
Electron capture internal conversion	Non-energetic electrons	0.0005	0.001	26
Beta minus particle	Energetic electrons	12	2.3	0.2
Alpha particle	Helium nuclei	0.1	9	80

**Table 2 molecules-26-02162-t002:** The recent status of PSMA-targeted alpha-particle-emitting radiopharmaceuticals.

Agent	Conjugator	Development Phase	Refs.
^213^Bi-J591	PSMA-targeting murine monoclonal antibody, J591	Preclinical study	[143,144,145]
^213^Bi-PSMA I&T	PSMA-targeting (3*S*,7*S*)-29,32-dibenzyl-5,13,20,28,31,34-hexaoxo-37-(4,7,10-tris(carboxymethyl)-1,4,7,10-tetraazacyclododecan-1-yl)-4,6,12,21,27,30,33-heptaazaheptatriacontane-1,3,7,26,37-pentacarboxylic acid (DOTAGA-FFK(Sub-KuE)), PSMA I&T	Preclinical study	[146]
^213^Bi-JVZ-008	PSMA-targeting nanobody, JVZ-008	Preclinical study	[146]
^211^At-6	PSMA-targeting (2*S*)-2-(3-(1-carboxy-5-(4-^211^At-astatobenzamido)pentyl)ureido)-pentanedioic acid, compound 6	Preclinical study	[147]
^225^Ac-PSMA-617	PSMA-targeting small molecule, PSMA-617	Clinical study	[127,138,141,148]
^227^Th-PSMA-TTC	PSMA-targeting fully human antibody, BAY 2315158	Clinical trial (phase I: NCT03724747)	[149]
^225^Ac-J591	PSMA-targeting murine monoclonal antibody, J591	Clinical trial (phase I: NCT03276572)	

## Data Availability

Not applicable.

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
