# Peer review of "Progress in Targeted Alpha-Particle-Emitting Radiopharmaceuticals as Treatments for Prostate Cancer Patients with Bone Metastases"

_molecules, 2021, doi:10.3390/molecules26082162_

Round 1

Reviewer 1 Report

In this manuscript the authors review the approaches for alpha particle therapy of bone metastases from prostate cancer.  In the Introduction, they have emphasized the importance of treating metastatic disease, elaborated the difference of targeting bone metastases from other organ metastases and described the use of [223Ra]RaCl2 for the treatment of bone metastases from prostate cancer.  The review encompasses 4 sections — 1) the biology of vicious cycle of bone metastases 2) current treatment options of bone metastases such as bisphosphonates denosumab and others 3) radiopharmaceuticals used for treating bone metastases and 4) radiopharmaceuticals that specifically target biomarkers present in the tumor within the bone metastatic environment.

This reviewer thinks that this is a well written manuscript covering the important aspects of alpha particle therapy of bone metastases from prostate cancer and it will be very useful for investigators working in this field. The authors should include a Table listing various approaches of alpha particle therapy including clinical trials that are currently used and those in pipeline.    

In addition, the authors need to address the following:

  1. Page 2, section 2: The first sentence is incomplete.
  2. Page 3, line 5: “converts” vs “convers”
  3. Page 8, line 7: “deprivation” vs “depravation”
  4. Novartis announced that the Phase III trial (VISION trial; NCT03511664) met the endpoints (SNMMI SmartBrief March 24, 2021). The authors may want to consider including this.
  5. Page 8, last para, line 7: “In their initial communication………” Whose communication? Need a reference.
  6. Page 9, 2nd para, 1st sentence: Provide a reference. Include the number of patients enrolled in this study.
  7. PSMA targeted agents labeled with other alpha emitters such as 213Bi, 227Th, 149Tb and 212Pb have been explored. This should be included with appropriate references. 
  8. The authors should include following references as well: 1) https://doi.org/10.1053/j.semnuclmed.2020.02.004 2) doi: 10.7150/thno.48107

Author Response

General Comment: In this manuscript the authors review the approaches for alpha particle therapy of bone metastases from prostate cancer.  In the Introduction, they have emphasized the importance of treating metastatic disease, elaborated the difference of targeting bone metastases from other organ metastases and described the use of [223Ra]RaCl2 for the treatment of bone metastases from prostate cancer.  The review encompasses 4 sections — 1) the biology of vicious cycle of bone metastases 2) current treatment options of bone metastases such as bisphosphonates denosumab and others 3) radiopharmaceuticals used for treating bone metastases and 4) radiopharmaceuticals that specifically target biomarkers present in the tumor within the bone metastatic environment.

Response: We thank the reviewer all the efforts to provide insightful comments. We believe that the quality of our manuscript has improved extensively by addressing the reviewer’s comments. We hope that the reviewers agree.

Comment 1: This reviewer thinks that this is a well written manuscript covering the important aspects of alpha particle therapy of bone metastases from prostate cancer and it will be very useful for investigators working in this field. The authors should include a Table listing various approaches of alpha particle therapy including clinical trials that are currently used and those in pipeline.   

Response: We thank the reviewer for pointing this out. We have provided this information in a new Table 2.

Minor Comments: In addition, the authors need to address the following:

  1. Page 2, section 2: The first sentence is incomplete.
  2. Page 3, line 5: “converts” vs “convers”
  3. Page 8, line 7: “deprivation” vs “depravation”
  4. Novartis announced that the Phase III trial (VISION trial; NCT03511664) met the endpoints (SNMMI SmartBrief March 24, 2021). The authors may want to consider including this.
  5. Page 8, last para, line 7: “In their initial communication………” Whose communication? Need a reference.
  6. Page 9, 2nd para, 1st sentence: Provide a reference. Include the number of patients enrolled in this study.
  7. PSMA targeted agents labeled with other alpha emitters such as 213Bi, 227Th, 149Tb and 212Pb have been explored. This should be included with appropriate references. 
  8. The authors should include following references as well: 1) https://doi.org/10.1053/j.semnuclmed.2020.02.004 2) doi: 10.7150/thno.48107

Response: All minor comments are now fixed accordingly.

Reviewer 2 Report

In this submission, the authors have provided an overview of the emergence of alpha particle therapies to the bone metastatic prostate cancer therapy space. The review covers areas that include historical bone targeted therapies, bone metabolism and turnover targeted Radium-223, emerging targeted alpha particle agents and a good deal of bone metastasis biology at the bone/cancer interface. Positives of this article include notably good references, fairly well written approach to describe some diverse scientific and clinical topics and an emphasis on some under-appreciated biology of metastasis (at least by the nuclear medicine field). This reviewer has two areas of concern: the first is minor English word choice, spelling and gammatical issues; the second is an incomplete coverage of relevant topics of safety and application.

For the minor issues, there are several areas where close rereading by the authors or perhaps an outside editorial service would be beneficial. Some examples include: the superfluous “-induced” in the title; the misspelling of “Raidopharmacuticals” in section headings 4+5; the nonsensical term non-energetic electrons”; etc. Another issue is the structure of the review. There are 6 sections, and two are nearly just paragraphs. It might greatly help the reader, and the organization of this paper, to have more subsections.

Some of the statements made in the manuscript are also an issue when placed under scrutiny. This is incorrect: "Importantly, to date, this is the only treatment modality that can prolong the survival time of cancer patients with bone metastases.” Hormone sensitive metastatic prostate cancer is commonly treated with hormone therapies, like ARN509, MDV3100 and other agents like rucaparib, have activity in bone metastatic prostate cancer. The comment you are attempting to make is perhaps that Radium-223 is the first effective *radiopharmaceutical* for the indication of bmCRPC. However, the blanket statement that you make misses much in the current prostate cancer patient treatment armamentarium.

This issue relating Radium-223 versus previous radiopharmaceuticals used for palliative care (as examples: 153Sm and 89Sr) is one that I think the authors must provide more information for. While there is some description of the differences in the physics of the particles that are emitted, there is little/no discussion of toxicities and dose limiting effects. Given enough 153Sm or 89Sr patients would likely benefit but the marrow toxicity from these longer ranged beta-particle therapies is a major limitation. In a review on alpha particles, it is necessary to describe in greater detail what kinds of toxicities may be avoided, and which may be incurred.

The other missing discussion topic is a lack of completeness when describing other alpha particle targeted radiopharmaceutical. Certainly, PSMA-617 is the most widely implemented radio ligand for prostate cancer at this time. However, it is certainly far from the only one in PSMA targeted therapies, nor is PSMA alone as a prostate cancer target. The authors need to provide more information on alternative ligands (including mini- and full antibodies) to PSMA; discuss PSMA-expression biology relating to bone metastasis versus soft tissue sites; as well as survey the current literature and clinical trial space with the goal of being more up to date for alpha particle therapies in the prostate cancer metastasis space. 

This review has the potential to be useful as a reference for the field, but more care needs to be taken in its editing and comprehensiveness.

Author Response

General Comment: In this submission, the authors have provided an overview of the emergence of alpha particle therapies to the bone metastatic prostate cancer therapy space. The review covers areas that include historical bone targeted therapies, bone metabolism and turnover targeted Radium-223, emerging targeted alpha particle agents and a good deal of bone metastasis biology at the bone/cancer interface. Positives of this article include notably good references, fairly well written approach to describe some diverse scientific and clinical topics and an emphasis on some under-appreciated biology of metastasis (at least by the nuclear medicine field). This reviewer has two areas of concern: the first is minor English word choice, spelling and gammatical issues; the second is an incomplete coverage of relevant topics of safety and application.

Response: We thank the reviewer all the efforts to provide insightful comments. We believe that the quality of our manuscript has improved extensively by addressing the reviewer’s comments. We hope that the reviewers agree.

Comment 1: For the minor issues, there are several areas where close rereading by the authors or perhaps an outside editorial service would be beneficial. Some examples include: the superfluous “-induced” in the title; the misspelling of “Raidopharmacuticals” in section headings 4+5; the nonsensical term non-energetic electrons”; etc. Another issue is the structure of the review. There are 6 sections, and two are nearly just paragraphs. It might greatly help the reader, and the organization of this paper, to have more subsections.

Response: We have strived to improve our grammatical accuracy and created subsections.

Comment 2: Some of the statements made in the manuscript are also an issue when placed under scrutiny. This is incorrect: "Importantly, to date, this is the only treatment modality that can prolong the survival time of cancer patients with bone metastases.” Hormone sensitive metastatic prostate cancer is commonly treated with hormone therapies, like ARN509, MDV3100 and other agents like rucaparib, have activity in bone metastatic prostate cancer. The comment you are attempting to make is perhaps that Radium-223 is the first effective *radiopharmaceutical* for the indication of bmCRPC. However, the blanket statement that you make misses much in the current prostate cancer patient treatment armamentarium.

Response: Sorry for the confusion. We are trying to point out that Rad-223 is the only bone-targeted treatment which extend the overall survival of bone metastatic patients (vs. bisphosphonate, denosumab). We have updated the information in our revised manuscript.

Comment 3: This issue relating Radium-223 versus previous radiopharmaceuticals used for palliative care (as examples: 153Sm and 89Sr) is one that I think the authors must provide more information for. While there is some description of the differences in the physics of the particles that are emitted, there is little/no discussion of toxicities and dose limiting effects. Given enough 153Sm or 89Sr patients would likely benefit but the marrow toxicity from these longer ranged beta-particle therapies is a major limitation. In a review on alpha particles, it is necessary to describe in greater detail what kinds of toxicities may be avoided, and which may be incurred.

Response: We have provided more information on beta-particle therapies, including discussion of toxicities and dose limiting effects.

Comment 4: The other missing discussion topic is a lack of completeness when describing other alpha particle targeted radiopharmaceutical. Certainly, PSMA-617 is the most widely implemented radio ligand for prostate cancer at this time. However, it is certainly far from the only one in PSMA targeted therapies, nor is PSMA alone as a prostate cancer target. The authors need to provide more information on alternative ligands (including mini- and full antibodies) to PSMA; discuss PSMA-expression biology relating to bone metastasis versus soft tissue sites; as well as survey the current literature and clinical trial space with the goal of being more up to date for alpha particle therapies in the prostate cancer metastasis space. 

Response: We have now provided this information in our revised text and in a new Table 2.

Comment 5: This review has the potential to be useful as a reference for the field, but more care needs to be taken in its editing and comprehensiveness.

Response: Again, we thank the reviewer all the efforts to provide insightful comments. We have strived to improve our grammatical accuracy and the quality of this manuscript.